# COVID-19 mRNA Vaccines Preserve Immunogenicity after Re-Freezing

**DOI:** 10.3390/vaccines10040594

**Published:** 2022-04-12

**Authors:** Santiago Grau, Elena Martín-García, Olivia Ferrández, Raquel Martín, Sonia Tejedor-Vaquero, Ramón Gimeno, Giuliana Magri, Rafael Maldonado

**Affiliations:** 1Pharmacy Department, Hospital del Mar, 08003 Barcelona, Spain; oferrandez@psmar.cat; 2Infectious Pathology and Antimicrobials Research Group (IPAR), Institut Hospital del Mar d’Investigacions Mediques (IMIM), 08003 Barcelona, Spain; 3Department of Medicine and Life Sciences (MELIS), Faculty of Health and Life Sciencies, Universitat Pompeu Fabra (UPF), 08003 Barcelona, Spain; 4Laboratory of Neuropharmacology-Neurophar, Department of Medicine and Life Sciences (MELIS), Universitat Pompeu Fabra (UPF), 08003 Barcelona, Spain; elena.martin@upf.edu (E.M.-G.); raquel.martin@upf.edu (R.M.); 5Neuroscience Research Program, Institut Hospital del Mar d’Investigacions Mediques (IMIM), 08003 Barcelona, Spain; 6Translational Clinical Research Program, Institut Hospital del Mar d’Investigacions Mediques (IMIM), 08003 Barcelona, Spain; stejedor@imim.es (S.T.-V.); rgimeno@imim.es (R.G.); gmagri@imim.es (G.M.)

**Keywords:** vaccination, COVID-19, mRNA, vaccine wastage, vaccine preservation, SARS-CoV2

## Abstract

The massive COVID-19 vaccine purchases made by high-income countries have resulted in important sample losses, mainly due to the complexity of their handling. Here, we evaluated the possibility of preserving the immunogenicity of COVID-19 mRNA vaccines after re-freezing vials, following the extraction of the maximum possible number of samples, as an alternative approach to minimizing their wastage. Thus, we exposed the vaccine vials to different re-freezing conditions and evaluated mRNA integrity and the effects in mice after in vivo administration. We reveal that the mRNA integrity of Comirnaty^®^ and Spikevax^®^ vaccines remained unaffected after re-freezing during 1 month at −20 °C or −80 °C. The immunological responses also remained unchanged in mice after these re-freezing conditions and no apparent side effects were revealed. The preservation of mRNA integrity and immunogenicity under these handling conditions opens the possibility of re-freezing the mRNA COVID-19 vaccine vials to limit their wastage and to facilitate vaccination processes.

## 1. Introduction

The advent of the COVID-19 pandemic induced the unusually rapid and effective development of various vaccines by the pharmaceutical industry that were first authorized and more recently approved by the main regulatory agencies. High-income countries bought large quantities of vaccines with the aim of quickly immunizing their citizens. The massive purchases of vaccines by these countries were also related to the high losses of such vaccines due to the complexity of their conservation and handling, as well as errors in the calculation of the real needs of the coverage of the population. This has not been the case for low- and middle-income countries, where the availability of vaccines is still scarce. Although several campaigns have been introduced for the distribution of vaccines to these countries, such as the Vaccines Global Access (COVAX) Facility [1], the results have been disappointing and far from reaching the initially set objectives [2]. The cost of the necessary infrastructure to ensure that 20% of the population of 92 low-income countries considered by this advanced market commitment receives the recommended vaccination schedule is estimated at USD 2 billion. However, excluding countries such as Kenya where more than 90% of the doses received through the COVAX program have been used, in most African countries, significant numbers of doses have been lost due to infrastructure problems which have prevented the implementation of an adequate vaccination program. It is currently estimated that the number of doses administered in Africa represents only 2% of the number of total administered doses in the world, with a coverage of 11.53% of the population in low-income countries compared to 67.89% of the population in high-income countries [3]. In addition, 240 million vaccine doses are estimated to have been lost in Japan and Europe [4].

The World Health Organization (WHO) has defined vaccine wastage as vaccines being discarded, lost, damaged or destroyed and highlights two processes which are mainly related with this wastage [5]. First, opened vials are usually discarded due to contamination during the handling process. With regard to unopened vials, vaccine wastage mainly occurs due to reaching the expiration date, storage at a higher than recommended temperature, or incorrect inventory. The presentation of vaccines in multi-dose vials has been underlined as one of the reasons for the loss of millions of vaccines as well as the cause of errors in the vaccination process [6]. The WHO has estimated at 2% the total loss of COVID-19 vaccines and underlined the main causes to be the alteration of storage temperature [7] as well as the donation of vaccines with an upcoming expiration date by high-income countries [8]. The experience of handling active vaccines against COVID-19 has led to changes in the recommendations for some of them with enhanced flexibility, mainly in terms of storage conditions [9]. However, simplifying the preparation of active vaccine doses against COVID-19 continues to be a challenge to avoiding unnecessary dose losses. Therefore, the introduction of strategies to improve vaccine handling could partially solve the vaccine wastage problem. 

Once Syringes with the vaccine doses have been prepared, they must be used just in time as part of a planned vaccination clinic without any possibility of re-freezing the leftover samples for later use. This represents a major limitation leading to a huge number of leftover vials being wasted since it is very difficult to predict the exact number of individuals that will be vaccinated in each clinic. The possibility of re-freezing reconstituted vials would be an enormous advance, allowing to maximize the potential utility of all reconstituted vials and minimize vaccine wastage.

However, scarce information is currently available about the consequences of re-freezing reconstituted vaccines in terms of integrity and the preservation of immunogenicity response. To the best of our knowledge, only one study has addressed this question by exposing vaccines to extreme freezing conditions (repeated freezing at temperatures reaching −250 °C) that are completely different to those used in vaccination centers but without testing the consequences of this in terms of the immunogenicity response [10].

The main objective of this study was to assess the consequences of re-freezing the remaining content of COVID-19 mRNA vaccine vials after the extraction of the maximum number of reconstituted vaccines as a possible alternative to minimize their wastage under same conditions as those used for human immunization. Therefore, the vaccine doses we administered to mice were chosen from previous studies that mimicked similar conditions extrapolated from human clinical practice for mouse vaccination with Cominarty^®^ [11] and Spikevax^®^ [12]. For this purpose, we exposed the remaining content of the COVID-19 mRNA vaccine vials under different re-freezing conditions and we evaluated the effects of mRNA integrity (Figure 1). These samples were also administered to rodents in vivo to evaluate the possible appearance of side effects and their ability to generate the immunological responses. 

## 2. Materials and Methods

### 2.1. Management of Vaccine Samples

Two brands of mRNA vaccines were used, namely Comirnaty^®^ (Biontech Manufacturing GMBH) and Spikevax^®^ (Moderna Biotech Spain S.L). All the vaccine samples used in this study were prepared in syringes from the remaining content of the vials from which all the possible doses for human vaccination were previously extracted. Therefore, all the vials were successfully used for human vaccination and none had been discarded for any other purpose. The conservation of these vials prior to their opening had been carried out in accordance with the specifications of the technical data provided by the supplier. For Comirnaty^®^, these specifications require a maximum period of 9 months between −90 °C and −60 °C and of 1 month between 2 °C and 8 °C once thawed. For Spikevax^®^, a maximum period of 9 months between −25 °C and −15 °C and of 30 days refrigerated between 2 °C and 8 °C and protected from light after thawing is required by the specifications.

The extraction of the remaining content of the vials and the preparation of the samples in the syringes was carried out in a horizontal laminar flow cabinet with the aim of guaranteeing a sterile and particle-free work environment. The syringes used (Becton Dickinson), made of polypropylene, have a capacity of 2 mL with three parts: body, plunger and sealing gasket. The extraction of the samples and their preparation in syringes was carried out within the period of stability of the vial considered in the data sheet: a total of 6 h between 2 °C and 30 °C after the first dose was withdrawn for Comirnaty^®^, and a total of 19 h between 2 °C and 25 °C for Spikevax^®^.

Samples extracted and prepared in syringes were exposed to three different experimental conditions before they were administered to mice (Figure 1):−A first group of samples was administered under the usual conditions reflected in the technical sheet. In this group, the administration of the samples to mice was carried out immediately after sample preparation under the usual conditions of vaccine administration in humans.−The syringes of a second group of samples were frozen at −20 °C during 1 month. The administration of the samples to the mice was carried out immediately after this re-freezing period, always preserving the expiration interval of the vaccine vial.−The syringes of a third group of samples were frozen at −80 °C during 1 month. The administration of the samples to the mice was also carried out immediately after this re-freezing period, always preserving the expiration interval of the vaccine vial.

For thawing, the frozen syringes were placed in the refrigerator on the same day that they were administered to the mice. Both refrigerators and freezers were connected to the Sirius^®^ System temperature and humidity monitoring system.

Mice were assigned to the different experimental groups and received the vaccine sample from the same assigned syringe.

### 2.2. Animals

A total of 70 Balb/c mice (35 male and 35 female), aged 2 months, were housed in groups of 4 mice per cage in temperature- and humidity-controlled laboratory conditions (21 ± 1 °C, 55 ± 10%) maintained with food and water ad libitum. Mice were tested during the light phase of the cycle (lights off at 8.00 p.m and on at 8.00 a.m). Mice were purchased from Janvier Labs (France). All experimental protocols were performed following the guidelines of the European Communities Council Directive 2010/63/EU and approved by the local ethical committee (Comitè Ètic d’Experimentació Animal-Parc de Recerca Biomèdica de Barcelona, CEEA-PRBB, agreement No. 11593). In agreement, maximal efforts were made to reduce the suffering and the number of mice used. 

### 2.3. Experimental Sequence in Animals

Mice were randomly distributed into seven groups (*n* = 10 per group with *n* = 5 females and *n* = 5 males in each group) and left to acclimate to the experimental room for 24 h. The seven groups of Balb/c mice were intramuscularly (IM) immunized via a needle injection using 10 µL Hamilton syringes either with 2 doses of 0.5 µg/Kg (1 µg/0.1 mL) of mRNA Comirnaty^®^ or Spikevax^®^ vaccines at 21 or 28 days interval, respectively (Figure 2A), or with saline solution (*n* = 10). Immunized mice were distributed according to different experimental conditions in groups that were IM injected with Comirnaty^®^ (1) under fresh conditions; (2) after storage at −20 °C; and (3) after storage at −80 °C, or in groups that received IM injection of Spikevax^®^ (4) under fresh conditions; (5) after storage at −20 °C; and (6) after storage at −80 °C. Before inoculation, mice were anesthetized with 5% isoflurane via inhalation in an induction chamber, and a maintenance dose of 1.5–2% was administered via a mask for IM inoculation. Blood samples were collected for IgG testing before receiving the 2nd dose or sacrifice, precisely at 20 or 27 days intervals for Comirnaty^®^ (day 20 and 41) or Spikevax^®^ (day 27 and 55). Blood extraction was performed by puncturing the submandibular facial vein using 5 mm Goldenrod lancets (BraintreeScientific, Inc), as previously described [13]. The security of immunization was assessed by ensuring that no alterations in body weight, general behavioral state of the mice using the Irwin test, or locomotor activity was produced. Body weight was measured three times during the procedure, before both IM immunizations and at the end of the experiment: at days 1, 21, and 41 for Comirnaty^®^ and at days 1, 28, and 55 for Spikevax^®^. The Irwin test was performed four times, at 24 h and 48 h after each vaccine administration: at days 2, 3, 22, and 23 for Comirnaty^®^ and 2, 3, 29, and 30 for Spikevax^®^. Finally, locomotor activity was assessed two times, precisely 24 h after each vaccine administration with Comirnaty^®^ at days 2 and 22 and Spikevax^®^ at days 2 and 29. 

### 2.4. Irwin Test

Irwin’s test is a battery of tests that assesses mouse behavior response after treatment with a compound and it is used to measure the security of the immunization of the mRNA vaccine. The test measures each mouse’s individual behavioral parameters in regularly spaced time intervals. The Irwin test was performed four times, at 24 h and 48 h after each vaccine administration, as described by Irwin [14]: at days 2, 3, 22, and 23 for Comirnaty^®^ and at days 2, 3, 29, and 30 for Spikevax^®^. 

### 2.5. Locomotor Activity

Locomotor activity was evaluated by using individual locomotor activity boxes (10.8 × 20.3 × 18.6 cm, Imetronic, Pessac, France) equipped with infrared sensors to detect locomotor activity and an infrared plane to detect rearings. The boxes were provided with a removable cage, a sliding floor, a trough, and a bottle. Mice were placed in the boxes for 1 h and the kinetics of the horizontal and vertical activity (the number of beam breaks) was recorded in blocks of 10 min. The locomotor activity test was performed at 24 h after each vaccine administration: at days 2 and 22 for Comirnaty^®^ and at days 2 and 29 for Spikevax^®^.

### 2.6. Analysis of mRNA Integrity

Microfluidic measurements to analyze mRNA integrity were performed using Agilent 2100 Bioanalyzer (Agilent Technologies, Santa Clara, CA, USA) with the RNA 6000 Nano LabChip kit and the assay Eukaryote total RNA Nano (Genomics Core Facility, University Pompeu Fabra, Spain), as previously described [15]. The results were generated and analyzed with the Bioanalyzer 2100 Expert Software (Version B.0210.SI764) and manual integration combined with smear analysis was used to define regions following the Bioanalyzer user guide. This technique has been validated to evaluate RNA integrity with high accuracy and precision [16], and it is widely used to determine such RNA integrity [17].

### 2.7. Production of Recombinant SARS-CoV-2 Spike RBD

The pCAGGS RBD construct, encoding for the receptor-binding domain of the wild-type SARS-CoV-2 Spike protein from the earliest lineage A virus (WT, YP_009724390.1, residues 319–541; NC_045512.2, A lineage) along with the signal peptide plus a hexahistidine tag was provided by Dr. Krammer (Mount Sinai School of Medicine, New York, NY, USA). RBD was expressed in-house in Expi293F human cells (Thermo Fisher Scientific) by the transfection of cells with purified DNA and polyethylenimine (PEI). Cells were harvested 3 days post-transfection and RBD-containing supernatants were collected by centrifugation at 13060 g for 15 min. RBD proteins were purified in Hitrap-ni Columns in an automated Fast Protein Liquid Chromatography (FPLC; Äkta avant), concentrated through 10 kDa Amicon centrifugal filter units (EMD Millipore), and resuspended in phosphate-buffered saline (PBS).

### 2.8. Sera Collection and Processing

Sera were collected from whole blood extracted from the tail vein 20 or 27 days post-prime and post-boost for mice immunized, respectively, with Comirnaty^®^ or Spikevax^®^ vaccine. Blood was incubated for 30 min without movement to trigger coagulation. Samples were then centrifuged for 5 min at 2000 g at room temperature and sera were collected and stored at −80 °C prior to use.

### 2.9. Enzyme-Linked Immunosorbent Assay (ELISA)

Ninety-six well flat bottoms high-bind plates (NUNC, Cat. 439454) were coated overnight at 4 °C with recombinant Spike RBD from SARS-CoV-2 original Wuhan strain at 1 µg/mL in PBS. Plates were washed with PBS 0.05% Tween 20 (PBS-T) and blocked with blocking buffer (PBS containing 1% bovine serum albumin, BSA) for 2 h at room temperature. Serum samples were serially diluted (starting dilution 1:30 and then 11 serial dilutions 1:3) in PBS-T supplemented with 1% BSA added to the plates and incubated for 2 h at room temperature. After washing, plates were incubated for 45 min at room temperature with horseradish peroxidase (HRP)-conjugated anti-mouse IgG secondary antibody (Southern Biotech, Cat.1030-05) diluted 1:4000 in PBS-T supplemented with 1% BSA. Plates were washed 5 times with PBS-T and developed by adding 90 µL of TMB substrate reagent set (BD bioscience). Then, the developing reaction was stopped with 90 µL 1M H2SO4. Absorbance was measured at 450 and 570 nm on a microplate reader (Infinite 200 PRO, Tecan). Optical density (OD) measurement was obtained after subtracting the absorbance at 570 nm from the absorbance at 450 nm. Negative threshold values were set using the mean of 1:30 dilution OD of PBS inoculated mice plus 4 times the standard deviation of the mean. To quantitate the level of RBD WT-specific IgG, sera end-point titers were calculated using Prism 8 (GraphPad) as the dilution that interpolated the sigmoid curve of each serum with the negative threshold.

### 2.10. Statistics

Behavioral data were analyzed by one-way ANOVA (group as a between-subjects factor) followed by multiple-group comparisons (Newman Keuls) when the main variable was significant using the Statistical Package for Social Science program SPSS 20.0. Results are expressed as mean ± SEM. 

GraphPad Prism (version 8.0) was used to conduct statistical analyses of the results obtained in immunogenicity studies. Two-tailed unpaired Mann–Whitney U test was performed to compare different vaccine treatments. Wilcoxon matched-pairs signed rank test was performed to compare time points in the same mouse. For each analysis, the type of statistical test, summary statistics, and levels of significance were specified in the figures and corresponding legends. 

All tests were performed two-sided with a nominal significance threshold of *p* < 0.05.

## 3. Results

### 3.1. Behavioral and Physiological Measurements

The potential incidence of behavioral and centrally mediated side effects of mRNA Comirnaty^®^ or Spikevax^®^ vaccine administration at 21 or 28 days interval, respectively (Figure 2A), was first evaluated using the Irwin test. The results showed an absence of behavioral and centrally mediated alterations after the administration of the Comirnaty^®^ or Spikevax^®^ vaccines and the scores obtained for all tested animals was equal to 0. Indeed, this 0 score was obtained in all the behavioral and central side effects evaluated in the Irwin test (tremor, Straub’s tail reaction, sedation, excitation, abnormal gait, jumping, loss of balance, motor incoordination, writhes, piloerection, stereotypes, head twitches, scratching, modified respiration, fear, aggressiveness, reactivity to touch, ptosis, exophthalmos, loss of traction, diarrhea, salivation, and lacrimation) after the administration of the first and the second dose of Comirnaty^®^ or Spikevax^®^ under the three experimental conditions evaluated (fresh, −20 °C, and −80 °C).

In agreement, the body weight gain was similar in all experimental groups. Indeed, statistical analysis revealed that the administration of the Comirnaty^®^ or Spikevax^®^ vaccines did not alter the body weight gain of mice across the experimental sequence. Specifically, the results showed the absence of significant differences between the groups (Figure 2B,C) and the expected progressive increase in body weight from the basal level of day 1 in all groups of mice tested (see Appendix A for statistical details). 

We also quantitatively evaluated the behavioral effects of different locomotor activity parameters, including horizontal and vertical activity measurements. No significant differences were obtained for the horizontal and vertical locomotor activity between the groups of mice tested, neither on day 2 nor on days 22 or 29 after the inoculation of Comirnaty^®^ or Spikevax^®^, respectively (Figure 2D–G and Appendix A). All groups showed a similar pattern of behavior, meaning that the administration of Comirnaty^®^ or Spikevax^®^ did not affect locomotor activity 24 h after the first or second vaccine administration. 

### 3.2. mRNA Integrity

The region of potentially degraded mRNA of Comirnaty^®^ or Spikevax^®^ vaccines was assessed in the samples used for the first or second dose comparing the experimental conditions of fresh, −20 °C and −80 °C conservation. Statistical results did not show significant differences between the groups of samples under any of the experimental conditions tested (Figure 3A–H and Figure 4A–H, and Appendix A). Thus, mRNA degradation was negligible in all samples, indicating that re-freezing the samples at −20 °C or −80 °C during 1 month did not alter mRNA integrity. In detail, all experimental conditions showed very low fluorescence unit (FU) values of less than 1 that corresponded to less than 5% of degradation over the total area for the Comirnaty vaccine and less than 6% of degradation for the Spikevax^®^ vaccine samples during both time-points of vaccine administration to mice. Representative figures of the RNA fractions area under the original mRNA peak of the Comirnaty^®^ or Spikevax^®^ vaccines samples of doses 1 and 2 are shown in Figure 3C–H and Figure 4C–H, respectively. 

### 3.3. Immunogenicity 

Immunogenicity was assessed by analyzing serum IgG to the receptor-binding domain (RBD) of the SARS-CoV-2 Spike protein by enzyme-linked immunosorbent assay (ELISA). As the RBD is part of the spike glycoprotein that mediates attachment to host cells and viral entry, RBD-specific IgG titers were previously shown to correlate with virus-neutralizing serum activity in humans and mice [18,19]. Sera were collected 20 or 27 days post-prime and post-boost for mice immunized, respectively, with Comirnaty^®^ or Spikevax^®^ vaccine. SARS-CoV-2 RBD-specific IgG was also analyzed in control mice (*n* = 10) inoculated with PBS alone.

As shown in Figure 5, the administration of the first dose induced the production of SARS-CoV-2 RBD-binding IgG for both mRNA vaccines, which further increased following the second dose regardless of the type of vaccine handling and storage conditions. Moreover, no significant differences were observed in SARS-CoV-2 Spike RBD-specific IgG titers among the three groups of mice after the first dose or post boost immunization with either Comirnaty^®^ or Spikevax^®^ vaccine. Of note, these results were confirmed even when male and female mice were analyzed separately (Figure 6).

## 4. Discussion

We first evaluated whether the administration of reconstituted mRNA COVID-19 vials handled under different conditions to those indicated in the technical specifications of the supplier may produce serious adverse events in mice. The administration of Comirnaty^®^ or Spikevax^®^ vaccine samples that were exposed to different handling and re-freezing conditions did not produce any behavioral nor physiological side effects in mice. This safety profile was initially evaluated using the Irwin test that contains standardized in vivo test batteries recommended by the International Council for Harmonisation [20] guidelines and it is routinely used in the pharmaceutical industry for pharmacological safety assessments to pre-clinically characterize drug-induced side effects before administration to humans [21]. The absence of behavioral and centrally mediated alterations in the Irwin test 24 h and 48 h after vaccine sample administration provides a first and important finding about the safety of these handled and re-frozen reconstituted mRNA vaccines. 

Another well-recognized and widely used pre-clinical tool to screen safety which is also often incorporated into safety regulatory submissions is the assessment of locomotor activity in rodents [22]. The automated system used in our approach allows a careful quantification of all locomotor parameters including horizontal and vertical activity [23] being highly sensitive to detect any possible alteration of these behavioral responses. No one of the locomotor parameters evaluated in our paradigm was modified by the administration of the vaccine samples exposed to different handling and re-freezing conditions further underlying the absence of major adverse events. 

Body weight gain and food intake were also similar during the whole experimental sequence in all groups of mice receiving COVID-19 vaccines handled and re-frozen under different conditions. In addition, all the animals of the different groups survived during the whole duration of the experiments excluding any possible early mortality associated with vaccine administration. The long period used for achieving the whole experimental sequence (41 days for Comirnaty^®^ and 55 days for Spikevax^®^) without any presence of major adverse events provides an additional safety argument. 

The different handling and re-freezing of the Comirnaty^®^ or Spikevax^®^ vaccines did not degrade the mRNA since the mRNA’s quality was similar under the three different experimental conditions evaluated. Indeed, the RNA fractions area under the original mRNA peak was similar under these conditions with very low values of FU that correspond to less than 5% of the original mRNA degradation for the Spikevax^®^ samples and less than 6% for the Comirnaty^®^ samples. Any possible product of the original mRNA’s degradation should be identified in these fractions of lower molecular weight. Although mRNA integrity remains unaffected after these conditions, the integrity of other components of the Comirnaty^®^ or Spikevax^®^ vaccines, such as the lipid nanoparticles that integrate the mRNA, could be potentially modified, which may compromise the immunogenicity obtained after in vivo administration. Nevertheless, mRNA degradation occurs at a faster rate than that of its vehicle and dictates the storage time and temperature of vaccines [24]. However, mRNA has been reported to be the most sensitive component in mRNA COVID-19 vaccines [25]. The strict handling and storage conditions required for these vaccines are due to the needs to preserve such an mRNA stability [25] which remained unaffected after handling and re-frozen Comirnaty^®^ and Spikevax^®^ vaccines under our experimental conditions. 

The most important correlate of protection of SARS-CoV-2 vaccines is the production of virus-binding and neutralizing antibodies, which therefore represent a major part in the evaluation of immunogenicity [26]. A strong positive correlation between SARS-CoV-2-neutralizing activity and RBD-specific IgG was reported in vaccinated individuals and in immunized mice [18,19]. Interestingly, Spike RBD-binding IgG titers evaluated in serum samples from mice immunized with the Comirnaty^®^ vaccine and exposed to different handling and re-freezing conditions were not significantly different. Spike RBD-specific-IgG titers were also comparable among mice vaccinated with Spikevax^®^ vaccines exposed under these different conditions, revealing that the immunogenicity obtained in mice was not modified by the exposure of both mRNA COVID-19 vaccines to these handling and re-freezing conditions. Since our studies were performed in male and female mice, additional studies could be performed in golden (Syrian) hamsters (*Mesocricetus auratus*) that would allow to have more accurate evaluation of safety [27,28]. A further assessment of a higher number of freeze and thaw cycles in the vaccine stability could be included in additional studies. 

Therefore, we demonstrated that reconstituted Comirnaty^®^ or Spikevax^®^ vaccines that have been handled and re-frozen at −20 °C or −80 °C preserve the integrity of mRNA and the immunogenicity responses obtained after in vivo administration in mice. No major side effects were revealed in mice that received these samples handled under different conditions to those indicated in the specifications of the supplier. The possibility of re-frozen the mRNA COVID-19 reconstituted vaccines in pre-filled syringes allows to overcome the limitations of infrastructures in middle- and low-income countries on vaccination campaigns as well as facilitate their distribution both to large cities as well as rural areas of these countries. Furthermore, the wastage of large quantities of vaccines observed in high-income countries [29] could also be minimized. Finally, the findings of this study could lead to changes in the recommendations of the storage and handling of the analyzed vaccines.

## Figures and Tables

**Figure 1 vaccines-10-00594-f001:**
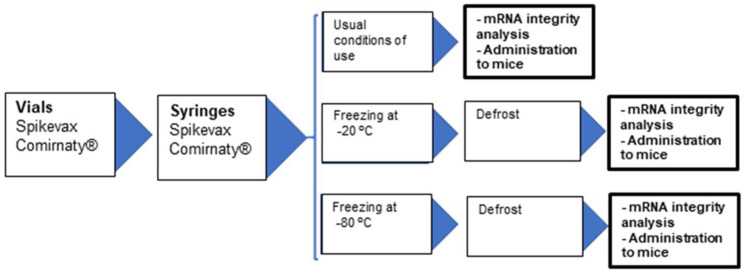
Summary of the protocol used for handling and re-freezing reconstituted Comirnaty^®^ and Spikevax^®^ COVID 19 vaccines.

**Figure 2 vaccines-10-00594-f002:**
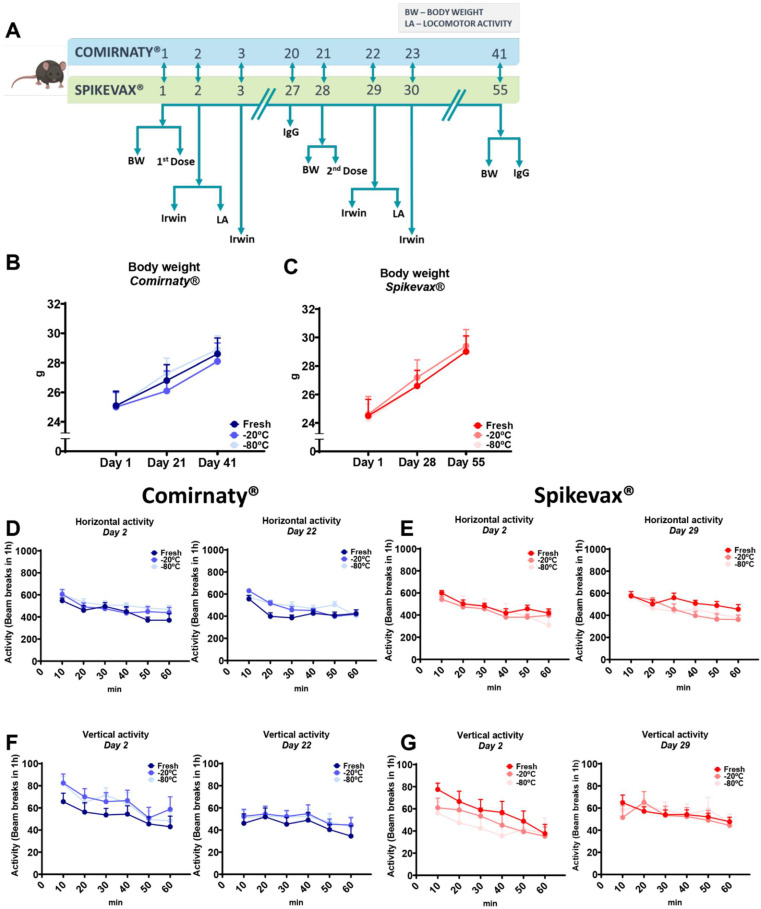
Different handling and re-freezing conditions of reconstituted Comirnaty^®^ and Spikevax^®^ COVID 19 vaccines did not affect behavioral and physiological measurements in mice. (**A**) Timeline of the experimental sequence in mice. (**B**,**C**) Bodyweight (BW) was measured three times during the procedure, at days 1, 21, and 41 for Comirnaty^®^ (**B**) and at days 1, 28, and 55 for Spikevax^®^ (**C**). (**D**–**G**) Locomotor activity measured in actimetry boxes. Beam breaks measured the horizontal and vertical activity for 60 min. Locomotor activity was assessed two times, precisely 24 h after each vaccine administration with Comirnaty^®^ at days 2 ((**D**)-left and (**F**)-left) and 22 ((**D**)-right and (**F**)-right) and Spikevax^®^ at days 2 ((**E**)-left and (**G**)-left) and 29 ((**E**)-right and (**G**)-right). All data are expressed in mean ± SEM (*n* = 10 for each group). Statistical details are included in Appendix A.

**Figure 3 vaccines-10-00594-f003:**
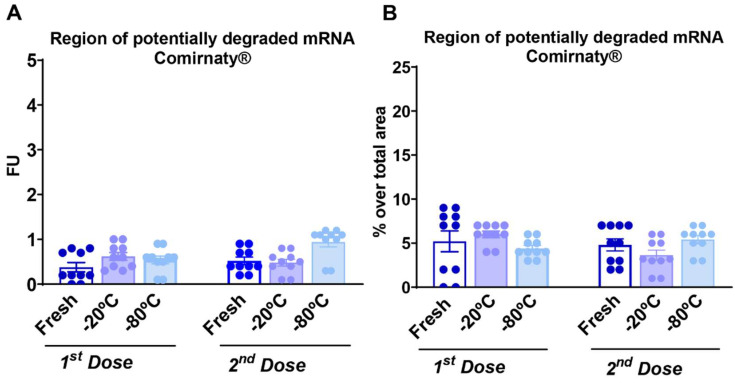
Different handling and re-freezing conditions did not affect mRNA integrity of reconstituted Comirnaty^®^ COVID 19 vaccines. (**A**) Measurement of the region of potentially degraded mRNA in fluorescence units (FUs) of the regularly used (fresh) samples or under −20 °C or −80 °C storage conditions in the first and second doses. Individual values of FU with the mean ± SEM are represented. (**B**) Percentage over the region’s total area of potentially degraded mRNA of the samples regularly used (fresh), or under −20 °C or −80 °C storage conditions in the first and second doses. Individual values of FU percentage with the mean ± SEM are represented (*n* = 10 per group). (**C**–**H**) Representative electropherograms expressed in FU of RNA integrity for different mRNA samples detailing the regions that indicate potentially degraded and intact mRNA peaks in the regularly used (fresh) samples or under −20 °C or −80 °C storage conditions in the first and second doses. Statistical details are included in Appendix A.

**Figure 4 vaccines-10-00594-f004:**
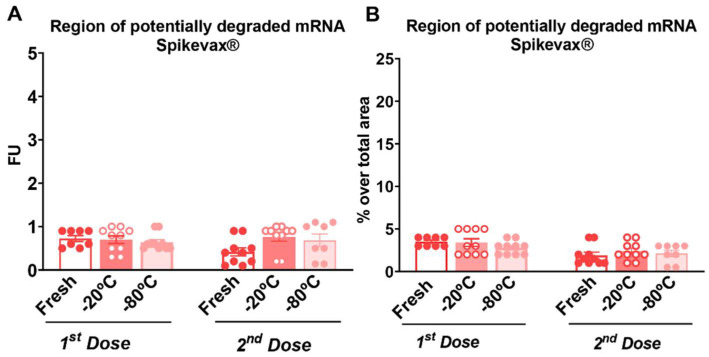
Different handling and re-freezing conditions did not affect mRNA integrity of reconstituted Spikevax^®^ COVID 19 vaccines. (**A**) Measurement of the region of potentially degraded mRNA in fluorescence units (FUs) of the regularly used (fresh) samples or under −20 °C or −80 °C storage conditions in the first and second doses. Individual values of FU with the mean ± SEM are represented. (**B**) Percentage over the region’s total area of potentially degraded mRNA of the samples regularly used (fresh), or under −20 °C or −80 °C storage conditions in the first and second doses. Individual values of FU percentage with the mean ± SEM are represented (*n* = 10 per group). (**C**–**H**) Representative electropherograms expressed in FU of RNA integrity for different mRNA samples detailing the regions that indicate potentially degraded and intact with mRNA peaks in the regularly used (fresh) samples or under −20 °C or −80 °C storage conditions in the first and second doses. Statistical details are included in Appendix A.

**Figure 5 vaccines-10-00594-f005:**
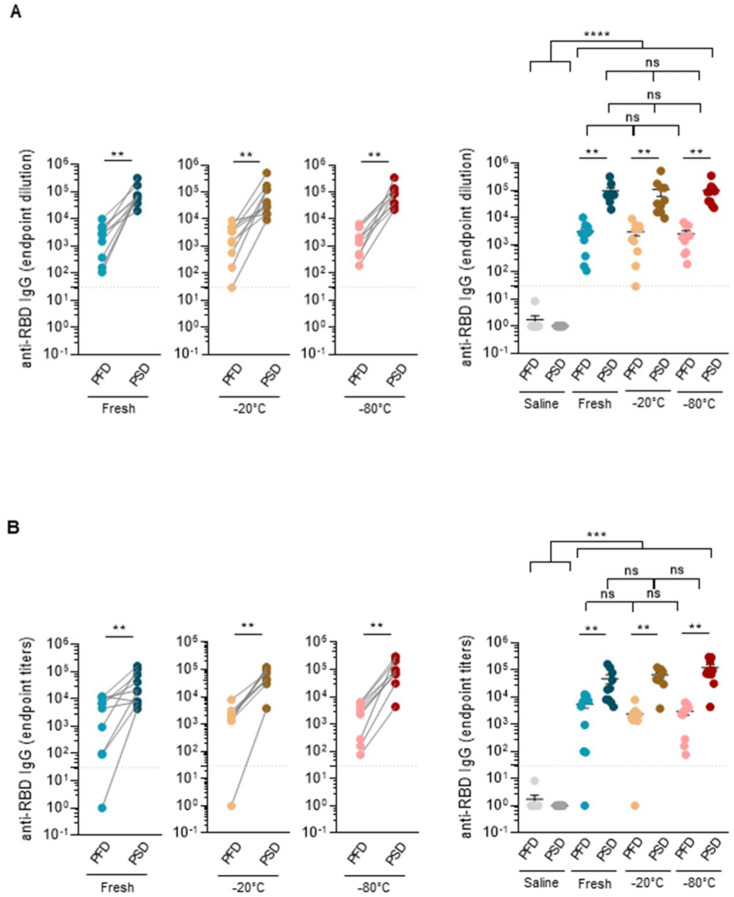
Different handling and re-freezing conditions did not affect the mRNA vaccine immunogenicity. SARS-CoV-2 Spike RBD-specific IgG end-point titers of mice immunized with Comirnaty^®^ (Pfizer-BioNTech) vaccine (**A**) or Spikevax^®^ Moderna vaccine (**B**) post-prime and post-boost immunization. Sera from the control mice inoculated with PBS were used to establish negative threshold values defined as the mean plus 4 times the standard deviation of the mean. Bars represent mean ± standard error mean (SEM). Dashed line indicates negative threshold. Data are presented as individual dots. Wilcoxon matched pairs test was performed to compare time points (post the first Dose, PFD; post the second dose, PSD). Two-tailed Mann–Whitney U test was performed to compare vaccine treatments (fresh, −20 °C and −80 °C) (ns = not significant, ** *p* < 0.01, *** *p* < 0.001 and **** *p* < 0.0001). For each time point and condition, *n* = 10 mice (5 males, 5 females).

**Figure 6 vaccines-10-00594-f006:**
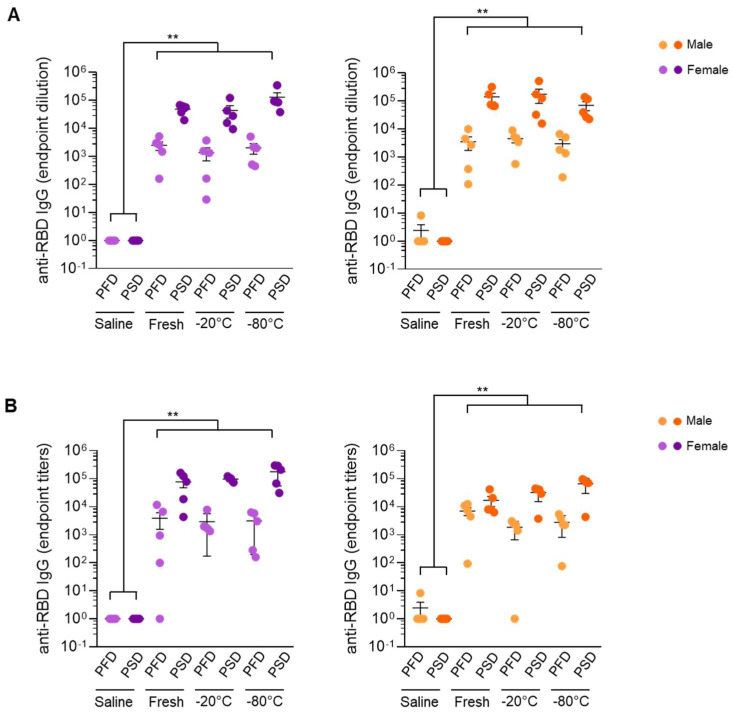
mRNA vaccines under different handling and re-freezing conditions induced similar responses in male and female mice. SARS-CoV-2 Spike RBD-specific IgG end-point titers of mice immunized with Comirnaty^®^ (Pfizer-BioNTech) vaccine (**A**) or Spikevax^®^ Moderna vaccine (**B**) post-prime and post-boost immunization. Sera from control mice inoculated with PBS were used to establish negative threshold values defined as the mean plus 4 times the standard deviation of the mean. Bars represent mean ± standard error mean (SEM). Dashed line indicates negative threshold. Data are presented as individual dots. Two-tailed Mann–Whitney U test was performed to compare vaccine treatment groups (saline, fresh, −20 °C, and −80 °C) (** *p* < 0.01). For each time point and condition, *n* = 5.

## Data Availability

The data that support the findings of this study are available from the corresponding author upon reasonable request.

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
