# Peer review of "COVID-19 mRNA Vaccines Preserve Immunogenicity after Re-Freezing"

_vaccines, 2022, doi:10.3390/vaccines10040594_

Round 1

Reviewer 1 Report

In this manuscript, the authors evaluated the possibility of preserving the immunogenicity of COVID-19 mRNA vaccines after re-frozen the vials. They demonstrated that mRNA integrity and immunological responses of vaccine remain unaffected after re-frozen during 1 month at -20ºC or -80ºC. This result does open the possibility of allowing mRNA COVID-19 vaccine vials to be re-frozen to limit their waste. However, there are still several issues which need to be addressed.

  1. the authors compared the immunological response between fresh and re-frozen mRNA vaccine. But they only tested one vaccine dose with two injections. The authors should also do a dose titration to compare the difference of immune response at low and high dose of vaccine. Thus they can confirm whether re-frozen could affect immunological response.
  2. in the method, does the author use (HRP)-conjugated anti-human IgG secondary antibody to test anti-RBD IgG in mouse serum? It’s completely not correct.
  3. in figure 6, the dot data about male mice used the unmatched color with legend.

Reviewer 2 Report

This manuscript addresses the rescue of overage in vaccine vials by pooling left over material, refreezing and storing at frozen temperatures prior to thawing and use. Overall, the authors present a solid piece of work that is well described. Some more detail could be added to the introduction on the importance or otherwise of utilizing residual drug product left in vials. It was not clear from the text how important this specific issue is deemed to be by authorities overseeing the administration of COVID-19 vaccines. Also, both freeze / thawing and storage temperature will be important. Some assessment of the number of freeze / thaw cycles these vaccines could tolerate would have added to the manuscript. This could at least be referred to based on available literature and potential future work.

Mice are not the preferred rodent model for studying the safety and immunogenicity of COVID-19 vaccines. The Syrian Hamster model is preferred. I am not suggesting the work needs to be conducted in hamsters for publication but there should be some reference to the need to conduct further safety studies in a better recognized model for COVID-19 prior to deploying the dose saving strategies suggested by this work.

Table 1 takes up quite a bit of space and its information could be conveyed in a couple of lines of text. Bottom line, there are no side effects. A table of this size is not necessary to convey this point. The detail on the tests conducted are adequately provided in the text already.

Typing size is a problem in Figure 2. Especially the axes. Also, in the text on lines 258-259, replace "Figure 2B-1C" with "Figure 2B-C".

The wording in section 3.2, lines 271-272 confused me. "Statistical results did not show non-significant differences between groups of samples in any of the experimental conditions tested." Should "non-significant" here be "significant". i.e. there were no significant differences. Even if correct as written, the double negative makes this confusing. Better to avoid the double negative and thereby not confuse the reader.

Type size on figures is even more of a case with Figures 3 and 4. The axes (scales) are unreadable. Also, it will greatly help the reader if the position of the specific peaks that correspond to intact versus degraded mRNA are pointed out on the Figure (if this is currently done it is unreadable).

A direct assay to assess virus neutralization activity would have been preferred, compared to receptor binding domain recognition. However, I think the assay provided is acceptable for this publication.

Good to see use of 10 animals per group in the mouse work, with an equal split of males and females.

On Figure 6, the color scheme seems to be off on the figure versus the legend (there are green dots on the legend but none on the figure: please check carefully).

It would have been interesting to look at other biophysical factors than just mRNA stability, such as liposome integrity and size? This could at least be referred to in the discussion.

Some attention to usage of the English language is needed. For example the repeated use of "frozen" rather than "freezing", as on line 17 in the abstract, "... after re-frozen the vials ..." should be "... after re-freezing the vials ...". There are several examples of this specific error and several other such English usage errors throughout the manuscript. Generally, it reads fine, but errors such as this crop up several times and tend to distract the reader, although the meaning remains clear. A few of the others that I noticed are replace "obtain" with "ensure" on line 40, and replace "used" with "use" on line 66, and replace "weigh" with "weight" on line 153. But I think the whole manuscript needs a careful read through for these kinds of error.

Round 2

Reviewer 1 Report

the authors have addressed my major concerns. there is no further question.